# Inherited Epigenetic Hallmarks of Childhood Obesity Derived from Prenatal Exposure to Obesogens

**DOI:** 10.3390/ijerph20064711

**Published:** 2023-03-07

**Authors:** María Á. Núñez-Sánchez, Almudena Jiménez-Méndez, María Suárez-Cortés, María A. Martínez-Sánchez, Manuel Sánchez-Solís, José E. Blanco-Carnero, Antonio J. Ruiz-Alcaraz, Bruno Ramos-Molina

**Affiliations:** 1Obesity and Metabolism Research Laboratory, Biomedical Research Institute of Murcia (IMIB), 30120 Murcia, Spain; 2Department of Obstetrics and Gynecology, ‘Virgen de la Arrixaca’ University Clinical Hospital, 30120 Murcia, Spain; 3Department of Nursing, Faculty of Nursing, University of Murcia, El Palmar, 30120 Murcia, Spain; 4Group of Pediatric Research, Biomedical Research Institute of Murcia (IMIB), 30120 Murcia, Spain; 5Respiratory and Allergy Units, Arrixaca Children’s University Hospital, University of Murcia, 30120 Murcia, Spain; 6Gynecology, Reproduction and Maternal-Fetal Medicine Research Group, Biomedical Research Institute of Murcia (IMIB), 30120 Murcia, Spain; 7Department of Biochemistry, Molecular Biology B and Immunology, School of Medicine, University of Murcia, Regional Campus of International Excellence “Campus Mare Nostrum”, 30100 Murcia, Spain

**Keywords:** obesogen, epiphenotypes, non-persistent organic pollutants, persistent organic pollutants, inorganic arsenic, childhood obesity, endocrine disrupting chemicals

## Abstract

Childhood obesity has reached epidemic levels in developed countries and is becoming a major cause for concern in the developing world. The causes of childhood obesity are complex and multifactorial, involving the interaction between individual genetics and environmental and developmental factors. Among the environmental factors, there is a growing interest in understanding the possible relationship between the so-called environmental obesogens and the development of obesity in children. Exposure to these obesogens such as phthalates, bisphenol A, or parabens, has been identified as a promoter of obesity through different mechanisms such as the alteration of adipocyte development from mesenchymal progenitors, the interference with hormone receptors, and induced inflammation. However, less attention has been paid to the inheritance of epigenetic modifications due to maternal exposure to these compounds during pregnancy. Thus, the aim of this review is to summarize the current knowledge of epigenetic modifications due to maternal exposure to those obesogens during pregnancy as well as their potential implication on long-term obesity development in the offspring and transgenerational inheritance of epiphenotypes.

## 1. Introduction

Childhood obesity has reached epidemic levels in developed countries and is becoming a major cause of social health concerns in the developing world. According to the World Health Organization (WHO), 39 million children under 5 years of age were overweight or obese in 2020. Currently, childhood obesity represents a public health challenge with a high economic burden associated with its treatment [1]. In fact, obese children are more likely to be obese in adulthood and are at a greater risk of developing long-term metabolic diseases including type 2 diabetes mellitus (T2DM), hypertension, non-alcoholic fatty liver disease (NAFLD), sleep apnea obstructive sleep (SAOS), and dyslipidemia [2]. Moreover, obesity has also been suggested to have a role in the development of other health conditions such as psychiatric- or bone-related diseases [3]. The causes of childhood obesity are complex and multifactorial, involving the interaction between individual genetics and environmental and developmental factors [4]. Indeed, it is known that the risk of developing chronic metabolic diseases, such as obesity, increases considerably in children born both to women who are obese at the beginning of pregnancy and to those with excessive weight gain during gestation [5,6].

In the past few decades, a growing body of evidence has shown that some environmental pollutants and chemicals are capable of promoting obesity, the so-called obesogens. Most of these types of pollutants (e.g., bisphenol A (BPA), phthalates, or dioxins) belong to the group of compounds known as endocrine disrupting chemicals (EDCs), which have been defined by the Endocrine Society as “exogenous chemicals, or mixture of chemicals, that interfere with any aspect of hormone action” [7] (Figure 1). Although the mechanisms of action are far from being completely understood, three of them are postulated as the most important mechanisms by which obesogens exert their effects: (i) altering adipocyte development; (ii) interfering with hormone receptors; and (iii) promoting inflammation. Obesogens are able to dysregulate adipocyte homeostasis through interacting with peroxisome proliferator-activated receptor gamma (PPARγ) and 9-cis receptor retinoic acid (RXR), and lead to an increased number and size of adipocytes, impaired glucose uptake and insulin signaling, reduced expression of brown adipocyte markers, and inflammation [8]. Furthermore, obesogens can indirectly promote obesity by altering the metabolic rate and energy homeostasis and disrupting the hypothalamic regulation of appetite and satiety [9]. They can also imitate the effects of natural estrogens by binding to their receptors (ERα and ERβ) and they interfere with androgen receptors (ARs) [10] and thyroid hormone receptors (TRs) [11], thus disrupting the endocrine system [12]. Current knowledge on molecular mechanisms underlying the action of obesogens has been recently reviewed elsewhere [13].

Although the genetic factors involved in the development of childhood obesity are only partially known, several studies have associated different epigenetic changes during the first months of life and the risk of obesity in the long term. Within these epigenetic changes, the analysis of changes in DNA methylation are the most studied in relation to increased adiposity [14]. In the case of adiposity in newborn children, these changes are mainly influenced by environmental factors related to the mother, such as obesity at the time of pregnancy or the lifestyle during pregnancy [15,16,17]. For example, methylation of the cg11531579 island (checkpoint with forkhead and ring finger domains genes, *CHFR*) site has been recently reported to be associated with rapid weight gain at early ages of life and the development of long-term obesity [14]. This agrees with the fetal origins of health and disease hypothesis, which states that the risk for the development of long-term diseases is associated with the events that occurred during early developmental periods, such as maternal exposure to environmental toxins [18]. For instance, it has been demonstrated that maternal smoking during pregnancy can lead to a decreased birthweight and an increased risk of obesity development during childhood [19]. Thus, due to their ability to promote epigenetic dysregulation on exposed individuals, obesogens have lately aroused the interest of researchers. Furthermore, prenatal exposure to these obesogens can cause epigenetic modifications that may produce adverse phenotypic outcomes in adulthood and may even be transferred to subsequent generations [20].

In this review, we will focus on the current knowledge on how prenatal exposure to obesogens induces epigenetic alterations that promote the development of obesity later in life (Figure 2), as well as the available information on the transgenerational effects associated with such exposure. For that purpose, we have focused on animal and epidemiological studies, which are summarized in Table 1, Table 2 and Table 3.

## 2. Non-Persistent Organic Pollutants

### 2.1. Bisphenol A

Bisphenol A (BPA) (4,4′-isopropylindenediphenol or 2,2′-bis(4-hydroxyphenyl)-propane) is a phenol compound that represents one of the most common compounds present in day-to-day life. It has been employed for the manufacture of polycarbonate plastics and epoxy resins found in several quotidian objects such as food containers, soaps, pipes, or car tires, among others [36]. Due to its short degradation half-life, it is not considered as a persistent organic pollutant but it is ubiquitous in the atmosphere, water, food, and dust. Consequently, human beings are exposed to high concentrations of this obesogen on a daily basis [37]. Indeed, several epidemiological studies have reported detectable BPA levels in human samples such as urine, umbilical cord blood, placenta, and breast milk [38]. BPA was one of the first xenoestrogen compounds identified and one of the best documented in relation to obesity incidence (recently reviewed elsewhere [39]). Thus, it is not surprising that BPA also represents the most studied EDC with regard to epigenetic alterations related to obesity development. Indeed, in utero exposure to low doses of BPA, it has been described to increase body weight, the adipocyte number, and abdominal fat in the offspring. All these changes are known to be associated with alterations in insulin, leptin, and adiponectin levels as well as in genes involved in adipogenesis, principally *PPARγ* expression, confirming the potential transferability of the obesogenic effect of BPA exposure [40,41].

The exposure of gestating females during the critical window of gonad sex determination leads to the transgenerational inheritance of altered epiphenotypes resulting from the dysregulation of the somatic cells of the adult even up to generation F5 [21,22,23,25]. In this regard, Jung et al. described that the administration of 50 mg/kg/day intraperitoneally (i.p.) to CD1 pregnant mice from embryonic day E7.5 to E13.5 was able to transfer the obesity epiphenotype up to the F5 generation. Furthermore, the authors showed that the increase in the body mass index (BMI) was correlated with increased food consumption and was conducted to alterations in sites containing binding motifs for the CCCTC-binding factor at two cis-regulatory elements (CREs) of the *Fto* (fat mass and obesity-associated) gene [21]. Another study carried out in CD1 mice administered BPA orally and showed that low (5 μg/kg body weight/day) but not high (500 μg/kg body weight/day) doses of prenatal exposure caused an increase in body weight and gonadal fat weight in F1 offspring in a sex-specific manner [23]. The screening of differential DNA methylation identified that this increase in body weight was correlated with the hypomethylation of the *Fggy* (carbohydrate kinase domain-containing) gene proximal to the transcription start site only in prenatal BPA-exposed males and increased expression levels of *Fggy* mRNA [23]. These sex-dependent effects of BPA were also observed in C57BL/6J mice where exposure to prenatal BPA increased body fat and perturbed glucose metabolism in male F1 and F2 offspring, which was in parallel to an increased DNA methylation at the *Igf2* (insulin-like growth factor 2) [24]. Similarly, the analysis of the sperm epigenome in the F3 generation showed alterations in the 197 DNA methylation regions including some obesity-associated genes including *Fgf19* (fibroblast growth factor 19)*, Esrra* (estrogen-related receptor alpha)*, Tnfrsf12a* (TNF receptor superfamily member 12A)*, Wnt10b* (Wnt family member 10B), and *Gdnf* (glial cell line-derived neurotrophic factor)*,* together with increased obesity incidence after prenatal exposure to a mixture of plastics (containing BPA, bis(2-ethylhexyl)phthalate, and dibutyl phthalate) at the lower dose employed [25].

Although BPA has been the most studied obesogen both in preclinical and clinical studies, there are only two clinical studies addressing the transgenerational inheritance of epiphenotypes in humans [22,33]. For instance, a recent study carried out using data from the Environment and Development of Children (EDC) study evaluated the effect of prenatal BPA exposure on obesity-associated CpG sites and early childhood BMI development [33]. The authors were able to establish a relationship between increased methylation at cg19196862 (*IGF2R*) and BMI at 2 years of age, although such an association was lost at later ages. Interestingly, when the analysis was performed separating by sex, the authors found that cg19196862 methylation correlated with an increased BMI at 4, 6, and 8 years in girls, but not in boys, which suggests a sex-specific effect of BPA exposure [33]. As the epigenome is known to be rapidly and constantly changing during the first years of life, it might be possible that epigenetic changes related to prenatal BPA-induced obesity development are given in earlier stages of life [42]. Hereof, it is worth highlighting the study carried out by Jungle et al., where they analyzed epigenetic alterations in the cord blood of 420 prenatally BPA-exposed children and the possible link to obesity development (part of the German prospective LINA mother–child cohort). The found correlations between hypomethylation of the obesity-associated mesoderm-specific transcript (*MEST*) promoter with increased *MEST* mRNA expression in the cord blood and high BPA prenatal exposure. The authors described that this increase in the *MEST* mRNA expression was associated with increased BMI z-scores at birth and at 6 years of age. These results were further confirmed in a mouse model of gestating BALB/c exposed to BPA (administered in drinking water), where the F1 generation of BPA-exposed mice had 53% higher fat mass than the controls and decreased methylation at the *Mest* promoter [22].

### 2.2. Phthalates

Phthalates, diesters of 1,2-benzedicarboxylic acid, are a class of obesogens that can be found in a multitude of products, including medications, sanitary products, toys, or food packaging [43]. Similarly to BPA, the exposure to phthalates occurs through ingestion, inhalation, or skin absorption and, although they have short biological half-live, humans are continuously exposed to them [44]. Phthalates can be divided into low molecular weight (LMW) phthalates (e.g., di-n-butyl phthalate (DBP)) and high molecular weight (HMW) phthalates, such as dimethyl hexyl phthalate (DEHP) or isononyl phthalate (DINP). LMW phthalates are hydrolyzed to monoesters and excreted, while HMW phthalates are further glucuronidated or sulfonated to be mostly excreted in urine. However, phthalate metabolites can also be found in feces, breast milk, saliva, or amniotic fluid, among others [43]. The mechanisms by which phthalates exert their obesogenic effect are similar to those described for BPA. Phthalates influence nuclear receptors (Erα, Erβ, AR, TR, and PPARγ receptors) and alter the regulation of energy homeostasis, leading to an altered glucose metabolism and increased risk of obesity development among other metabolic abnormalities [45].

Epidemiological and animal studies have shown that phthalates are related to the occurrence of metabolic disorders such as obesity and T2DM [46,47,48,49]. Similarly, prenatal exposure to phthalates has also been associated with obesity development in the offspring [50,51,52], although studies on the epigenetic inheritance of obesity are still scarce. For instance, the use of the yellow agouti mouse model has shown that offspring that were prenatally exposed to both individual or different mixtures of phthalates (DEHP, DBP, and DINP) had increased body weights and altered DNA methylation at intracisternal A-particles in the F1 generation [26]. Furthermore, the alterations of body weight after exposure to the different phthalate treatments were dependent on the sex and genotype [26]. On the other hand, exposure to a mixture of BPA and phthalates increased body weight in both female and male F3 generations and was related to alterations in DNA methylation in obesity-related genes [25]. Regarding epidemiological studies, there is only one addressing the transgenerational inheritance of obesity-related epigenetic alterations. Recently, Miura et al. analyzed the levels of the DEHP primary metabolite mono(2-theylhexyl)phthalate concentration (MEHP) (as an indicator of DEHP exposure) in maternal blood samples and identified increased methylation levels in genes related to metabolism. Moreover, the increase in methylation levels at CpGs cg27433759 (*PIK3CG*, phosphatidylinositol-4,5-bisphosphate 3-kinase catalytic subunit gamma), cg10548708 (*ACAA1*, acetyl-coA acyltransferase 1), and cg07002201 (*FUT9*, fucosyltransferase) were found to be associated with higher levels of MEHP prenatal exposure and lower ponderal index at birth [34].

## 3. Persistent Organic Pollutants

### 3.1. Tributyltin

Tributyltin (TBT) belongs to the organotin family, which are tin compounds (R(_4−n_)SnX_n_) covalently bonded to one or more organic chains and another functional group [53]. Due to its biocidal properties, TBT has been given multiple uses in different industry sectors such as fungicide, acaricide, wood preservative, and antifouling paints, and it represents the most common contaminant of both marine and freshwater ecosystems [53,54,55]. It is considered a persistent organic pollutant with a slow rate of environmental degradation that varies from days to months in water, and up to several years in sediments [56]. In the case of humans, exposure to TBT occurs in multiples ways such as atmospheric contamination, or the consumption of contaminated seafood, water, and beverages [57].

The obesogenic effect of TBT has been described to be via PPARγ and the retinoid X receptor (RXR) activation leading to the promotion of adipogenesis and lipogenesis [58,59]. In vivo studies have described that TBT exposure induces adiposity and lipid accumulation in the liver, and perigonadal WAT in males and females [60,61]. Furthermore, these alterations have been found to be transmitted transgenerationally [62]. For instance, Chamorro-García et al. showed that prenatal TBT exposure increased white adipose tissue depot weights, adipocyte size, and number and intrahepatic lipid accumulation in mice at 8 weeks of life, which was transferred to the non-exposed subsequent generations [62]. Further studies from the same group have revealed that these alterations in body weight mainly affected the male descendants (F2 to F4 generation) and were related to altered chromatin organization, leading to changes in DNA methylation and the expression of genes involved in energy metabolism such as *Lepr* (leptin receptor) or *Apoc4* (apolipoprotein C4) [27,28,63]. Moreover, an RNA-seq analysis showed that these inherited transgenerational traits not only affected the DNA methylation status or chromatin organization but also the non-coding RNA. In this regard, it has been described that the dysregulation of lncRNA (long non-coding RNA) involved in glucose homeostasis such as Gm6277 or Gm10804 as well as in adipose accumulation (Rian) can persist up to the F4 male generation [64]. Similarly, miR-223, a key mediator of PPARγ-dependent macrophage alternative activation, was found to be overexpressed in F2 and F3 male descendants of TBT-exposed dams [65].

### 3.2. Parabens

Parabens are alkyl esters of p-hydroxybenzoic acid with high antimicrobial activity that are widely employed in cosmetics, personal care, and the pharmaceutical industry [66]. Parabens are known endocrine disruptors that are able to activate PPARs, TRs, and glucocorticoid receptors (GRs). In a recent review on the obesogenic effect of prenatal exposure to parabens, Xu et al. [67] concluded that despite the evidence from in vitro and animal studies, there are several discrepancies regarding population-based studies, which are probably due to the inherent limitations of the studies. Interestingly, it is worth highlighting that although most of these studies confer an obesogenic effect to most parabens (butylparaben, methylparaben, or propylparaben), prenatal exposure to ethylparaben (EtP) has been described to be associated with lower birth weights and lower BMI z-scores [35,68]. In this regard, increased EtP placental concentrations have been correlated to hypermethylation of cg08612779 in humans (annotated to gamma-glutamylransferase 7 (*GGT7*)) [35]. GGT7 is a hepatic enzyme involved in the metabolism of glutathione and in the transpeptidation of amino acids known to be upregulated in different metabolic disorders [69]. However, cord blood biomarker analysis showed contradictory results, as the EtP levels were positively associated with GGT, leading the authors to be cautious in establishing a causal link [35]. On the other hand, prenatal exposure to butylparaben (BuP) has been described to increase BMI during the first 8 years of life, especially in girls. Here, the possible epigenetic alterations were not explored in the clinical trial, but a deeper mechanistic evaluation in mice showed DNA hypermethylation in the regulatory regions of the proopiomelacortin (*Pomc*) gene in the hypothalamus of BuP-exposed offspring, thus suggesting that the observed obesity phenotype might be driven by alterations in the neuronal regulation of satiety and hunger [30].

### 3.3. Dichlorodiphenyltrichloroethane (DDT)

DDT (1,1,1-Trichloro-2,2’bis(p-chlorophenyl) ethane) was introduced as a pesticide in the 1940s and was extensively used until the mid-1960s [70]. Despite its use being generalized banned since the 1970s, DDT remains to be used in vector control [71]. Furthermore, due to its chemical stability and the long half-life of DDT and its metabolites (mainly 1,1-dichloro-2,2-bis(p-chlorophenyl)ethylene (DDE)), the implications of DDT exposure to human health remain an important question nowadays. Over the last four decades, epidemiological studies have related DDT exposure to multiple disorders such as cardiovascular disease [72], cancer [73], T2DM [74], and obesity [75]. Indeed, the potential endocrine-disrupting and obesogenic properties of DDT and its metabolites have gained interest over the last decade. A recent systematic review and meta-analysis performed by Stratakis et al. [76] described a positive association of prenatal exposure to DDE with the BMI z-score in children from 2 to 9 years of age. On the other hand, the analysis including children from 0 to 2 years of age showed no correlation. As these years in early life represents a critical stage where most reprograming events take place, it is reasonable to think that among the mechanisms involved in early obesity development, epigenetic changes may have a key role. In this regard, the exposure of gestating rats to relevant levels of DDT has been shown to have no effect on body weight in the first offspring generation. However, after three generations, obesity incidence increased by 50% and was correlated with the presence of 39 differentially methylated regions (DMR) with low CpG regions [29]. Similar results were found in the F4 generation and with other pesticides and a variety of EDC such as dioxins by the same authors [77,78,79,80,81,82].

### 3.4. Polycyclic Aromatic Hydrocarbons

Emerging evidence has suggested that polycyclic aromatic hydrocarbons (PAHs), products from the incomplete combustion of organic materials, may also contribute to the development of metabolic diseases, including T2DM and obesity [83,84]. In particular, hydroxylated PAH metabolites are able to activate ERs causing endocrine disruptive effects in different organs [85]. Previous studies have shown that prenatal PAH exposure altered DNA methylation in several genes that correlated to a greater asthma occurrence by 5 years of age [86]. Although prenatal PAH exposure has been related to childhood obesity development [87,88], mechanistic studies are still limited. In this regard, Yan et al. exposed pregnant BALB/cByj pregnant dams to a mixture of PAHs via nebulizers from GD1 to GD19. As expected, the F1 offspring from the PAH-exposed dams exhibited an increased body weight, gonadal WAT, WAT and BAT adipocyte size, and fat mass of inguinal WAT. Moreover, these effects were also observed in the grand-offspring mice. Further evaluation of the possible mechanisms demonstrated that these alterations in body weight and body fat were associated with the decreased DNA methylation of the *Pparγ* promoter at different CpGs, which might be responsible for such effects, although a marked sexual dimorphism was observed [31].

## 4. Inorganic Arsenic

Inorganic arsenic (iAs) is one of the heavy metals currently recognized as an endocrine disruptor. Exposure to iAs occurs mainly via drinking water and it is naturally found in soil and groundwater in many countries worldwide [89]. Long-term exposure to iAs can lead to the development of several health conditions such as skin lesions, peripheral neuropathy, cancer, or diabetes [90]. In relation to obesity, iAs has shown unconclusive results in both epidemiological and animal studies, and the potential mechanisms are not entirely understood [91,92]. However, because iAs can cross the placental barrier, it has been suggested that it could cause several physiological and epigenetic changes that may contribute to obesity development in the offspring [93]. A recent animal study using C57BL6/J mice has shown that maternal exposure to relevant iAs concentrations induced alterations in the wean weight and weight gain in F1 and F2 offspring in a sex- and dose-specific manner [32]. Furthermore, these alterations were also generation-dependent, as while the F1 generation from dams exposed to iAs showed a reduced wean weight, it was increased in the F2 offspring. Interestingly, the analysis of epigenetic modifications in both generations showed that only a few differentially methylated CpG sites (DMCs)-containing genes were common between F1 and F2, and no DMR-containing genes were shared intergenerationally, supporting the sex- and generation-specific effects observed after prenatal iAs [32].

## 5. Future Perspective and Conclusions

The accumulating evidence strongly supports the role of several pollutants in the development of metabolic-related pathologies, including obesity. In this regard, we can conclude that the effect of pollutants upon the epigenetic characteristics of the population and the heritability of such potential modifications, which may lead to the development of obesity and related diseases, is of great interest nowadays for industrialized societies in which obesity and its comorbidities are becoming a remarkable health burden.

The epigenetic effect of obesogenic compounds such as those herein listed has been mainly limited to studies performed in animal models. However, although the use of these models is fundamental to understanding the mechanisms underlying the transgenerational inheritance of epiphenotypes, the results should be extrapolated with caution in relation to humans. Thus, it is of great interest to increase the number of long-term human studies that would provide new data and shed some light on the obesogenic effect of pollutants upon an offspring’s metabolic development. Therefore, another conclusion is that additional longitudinal studies should be performed not only in large mother–offspring cohorts but also in father–offspring cohorts to study epigenetic modifications caused by obesogens, with a special focus on the time frame including pregnancy and the first 2 years of life after birth.

## Figures and Tables

**Figure 1 ijerph-20-04711-f001:**
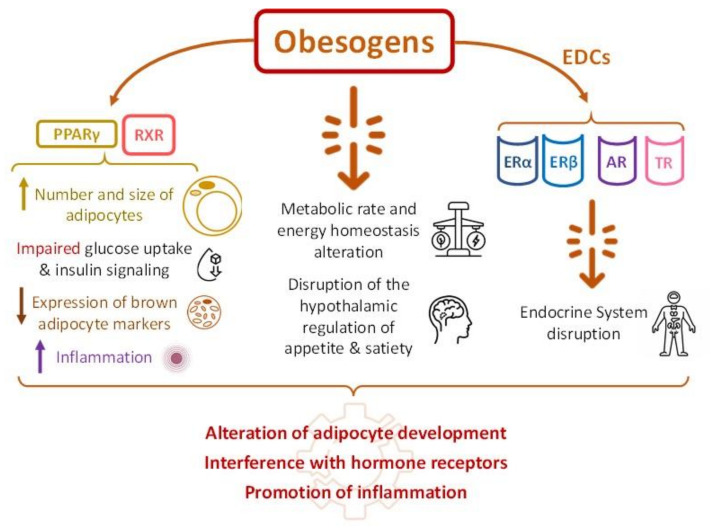
Mechanisms mediating the health-disruption effects of obesogens and the corresponding alterations provoked. AR, androgen receptor; EDCs, endocrine disrupting chemicals; ERα, estrogen receptor alpha; ERβ, estrogen receptor beta; PPARγ, proliferator-activated receptor gamma; RXR, 9-cis receptor retinoic acid; and TR, thyroid hormone receptor.

**Figure 2 ijerph-20-04711-f002:**
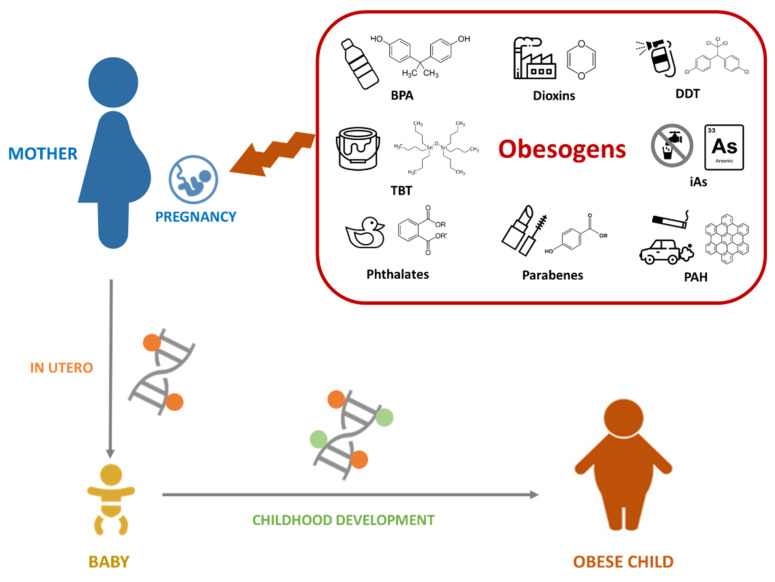
Schematic effect of obesogens on epigenetic hallmarks leading to childhood obesity. Several environmental pollutants may alter the inherited epigenetic profiles during pregnancy, which would contribute to the later development of childhood obesity and other metabolic related diseases. BPA, bisphenol A; DDT, dichlorodiphenyltrichloroethane; iAs, inorganic arsenic; PAH, polycyclic aromatic hydrocarbons; and TBT, tributyltin.

**Table 1 ijerph-20-04711-t001:** Animal studies on transgenerational inheritance of obesity epiphenotypes after ancestral exposure to obesogenic non-persistent pollutants.

Compound	Model	Treatment	Outcomes in Progeny	Reference
Bisphenol A (BPA)	CD1 mice	Dams administered 50 mg/kg/day i.p. of BPA or sesame oil daily to gestating F0 CD1 females from embryonic day E7.5 to E13.5.	↑ 60% and 97% obesity in the male F2 and F4 progeny, respectively. Obesity phenotype transmittable up to F6. ↑ Visceral WAT and adipocyte size. ↑ Circulating leptin levels. ↑ Food intake. Altered light/dark respiratory exchange ratio. Demethylation in a cis-regulatory element of the *Fto* gene.	[21]
Bisphenol A (BPA)	BALB/c mice	Dams exposed to BPA (5 μL of BPA/mL of drinking water) 1 week before mating until delivery of the offspring.	↑ BW and fat mass. ↓ *Mest* methylation with increased *Mest* mRNA expression.	[22]
Bisphenol A (BPA)	CD-1 mice	Pregnant CD-1 mice (F0) dosed orally with 0, 5 (low-BPA group), or 500 (high-BPA group) μg/kg/day of BPA in tocopherol-stripped corn oil or vehicle during gestational days 9 to 18.	↑ Whole BW in males F1 (low-BPA group). ↑ Gonadal fat in F1 males (low-BPA group). ↓ Methylation of *Fggy* promoter in F1 males (low-BPA group). ↑ *Fggy* mRNA expression in F1 males (low-BPA group). No differences in F1 female mice between the groups.	[23]
Bisphenol A (BPA)	C57BL/6 mice	Female C57BL/6 mice exposed to BPA at 50 μg/kg food (low-BPA group), BPA 50 mg/kg food (high-BPA group), or vehicle (7% corn oil) from two weeks before mating until weaning.	↓ Birth weight in F1 male low-BPA group. Accelerated weight gain in F1 male low-BPA group. ↑ BW and fat content at between postnatal day 98 and 117 in F1 male BPA groups. ↑ Insulin levels in F1 male BPA groups. ↑ DNA methylation at the *Igf2* differentially methylated region (DMR) 1 in F1 and F2.	[24]
Bisphenol A (BPA), diethylhexyl phthalate (DEHP) and dibutyl phthalate (DBP)	Sprague Dawley rats	Daily injection of either DMSO or mixture of BPA, DEHP, and DBP at high dose (50, 750, and 66 mg/kg/day, respectively) or low dose (25, 375, and 33 mg/kg/day, respectively) during embryonic days 8 to 14 of gestation.	↑ BW in the F3 males low-dose plastic lineage. ↑ BW and abdominal fat deposition in the F3 female high- and low-dose plastics lineage. Transgenerational sperm epigenome alterations in 197 DMRs some of which are associated with *TNFRSF12A*, *ESRRA*, *FGF19*, *WNT10B*, and *GDNF* obesity genes.	[25]
Diethylhexyl phthalate (DEHP), diisononyl phthalate (DINP), and dibutyl phthalate (DBP)	Yellow agouti (Avy) mice	Virgin dams at 6–8 weeks consuming 7% corn oil (control), DEHP 25 mg/kg, DBP 25 mg/kg, DINP 75 mg/kg chow, 25 mg DEHP + 75 mg DINP/kg chow or 25 mg DEHP + 75 mg DINP + 25 mg DBP/kg chow from 2 weeks before mating until weaning.	↑ BW of female F1 mice from F0 mice exposed to DINP or the mixture of the three phthalates. ↑ BW of male F1 mice from F0 mice exposed to DEHP, DINP, and DEHP + DINP. ↑ Liver weight in exposed female offspring exposed to DINP, DEHP + DINP, and DEHP + DINP + DBP. ↑ Gonadal fat weight female offspring exposed to DEHP + DINP + DBP. ↓ Intracisternal A-particle (IAP) DNA methylation in males. ↑ IAP DNA methylation in females.	[26]

BPA, bisphenol A; BW, body weight; DBP, dibutyl phthalate; DEHP, diethylhexyl phthalate; DINP, diisononyl phthalate; DMR, differentially methylated regions; DMSO, dimethyl sulfoxide; *ESRRA*, estrogen related receptor alpha; *FGF19*, fibroblast growth factor 19; *Fggy*, carbohydrate kinase domain-containing; *GDNF*, glial cell derived neurotrophic factor; i.p., intraperitoneal; IAP; intracisternal A-particle; *Mest*, mesoderm-specific transcript; *TNFRSF12A*, TNF receptor superfamily member 12A; WAT, white adipose tissue; and *WNT10B*, Wnt family member 10B; ↑, increase; ↓ decrease.

**Table 2 ijerph-20-04711-t002:** Animal studies on transgenerational inheritance of obesity epiphenotypes after ancestral exposure to obesogenic persistent pollutants.

Compound	Model	Treatment	Outcomes in Progeny	Reference
Tributyltin (TBT)	C57BL/6J mice	Female C57BL/6J mice exposed to either 50 nM TBT or 0.1% DMSO via drinking water 7 days before mating and continued throughout lactation. F4 mice fed with standard diet (STD) until 19 weeks of age and then with either chow or high-fat diet (HFD) for 6 weeks and then STD diet until 33 weeks of age.	↑ BW in F1 females. ↑ Gonadal WAT content in F2-F4 male. ↓ Inguinal adipose depot in males. No differences in BW between the groups. ↑ Body fat in F4 males after exposure to HFD. ↓ Fat mobilization in F4-TBT males. ↑ Leptin levels at week 33 in F4-TBT males. Hypermethylation of gonadal WAT genome associated with changes in expression of genes involved in metabolic processes.	[27]
Tributyltin (TBT)	Transgenic OG2 mice	Female OG2 mice exposed to either 50 nM TBT or 0.1% DMSO via drinking water 7 days before mating and continued throughout lactation until 3 weeks after delivery. F3 mice fed with chow diet until 19 weeks of age, with either chow or HFD for 6 weeks.	↑ Fat mass in TBT-F3 males after HFD. ↓ Body fluids and lean mass with no increase in BW in TBT-F3 males after HFD. Presence of normal-weight obesity syndrome. No transgenerationally persistent changes in CpG methylation promoter-associated CpG islands in fetal testes. Chromatin compaction protein MORC1 led to RNA expression changes in representative mouse endogenous retroviruses.	[28]
Dichlorodiphenyltrichloroethane (DDT)	Hsd: Sprague Dawley^®^™SD^®^™ rats	Pregnant rats administered either i.p. 50 or 25 mg/kg BW/d of DDT or vehicle daily from gestation days G8 to G14.	No changes in BW in the F1 generation. ↑ Obesity incidence in the F3 generation by 50%. Transgenerational transmission of disease through both female and male germlines. ↑ Testis, ovary, and kidney abnormalities in F1 and F3 generations. Identification of 39 differential DNA methylated regions (DMR) in sperm from F3 DTT lineage. ↓ Density CpG regions (CpG desert).	[29]
Parabens	BALB/cByJ mice	Pregnant dams exposed subcutaneously to 1.75 μg of butylparaben (BuP) in 100 μL corn oil or vehicle twice a week until weaning.	↑ BW, fat mass, food intake fasting serum glucose levels, and adipocyte size in female offspring from BuP-exposed dams. ↓ Lean mass in female. No changes in males in any of the parameters measured. ↓ POMC expression via hypermethylation of mPE1 regulatory region of *Pomc*.	[30]
Polycyclic aromatic hydrocarbon (PAH) mixture.	BALB/cByj mice	Female BALB/cByj mice exposed to either control aerosol solution (99.97% water, 0.02% Tween 80, and 0.01% antifoam) or 7.29 ng/m3 PAH mixture (3.69 ng/m3 pyrene plus 3.60 ng/m3 from eight other individual PAH) administered via nebulizers from gestational day (GD) 1-3 through GD 19-21 or until day of delivery.	↑ BW on PND25-27 and PND52-60 for females and PND30-60 for males. ↑ Fat mass of inguinal WAT in PAH female offspring on PND60. ↑ Gonadal WAT in PAH-offspring at PND60. ↑ WAT and BAT adipocyte size in PAH group. ↑ *Pparγ*, *Cox2,* and *Adiponectin* mRNA expression in WAT and BAT PAH-offspring. ↓ *Fas* expression in WAT and BAT in male PAH-offspring. ↓ DNA methylation of *Pparγ* promoter at CpG-303 in female and CpG-303 and CpG-189 in male offspring in inguinal WAT. ↓ DNA methylation of *Pparγ* promoter at CpG-303 and CpG-195 in female and CpG-303 in male offspring in interscapular BAT.	[31]
Inorganic arsenic (iAs)	C57BL/6J mice	F0 female mice exposed to 1 (control, tap water), 10, 245, or 2300 ppb of inorganic arsenic (iAs) in drinking water two weeks before mating until delivery.	Sex- and dose-specific differences in weight and body composition in F1 and F2 generations. ↓ Wean weight in F1 generation from dams exposed to 10 and 2300 ppb iAs. Altered glucose metabolism in F1 female offspring of 10 ppb. ↑ Wean weight in 10 ppb F2 male offspring. ↓ Weight gain and final body mass to fat mass ratio in F2 male in all treatments. ↑ Differentially methylated CpGs (DMC) and DMR in F2 female generation exposed to 10 and 245 ppb iAs. No DMR shared between F1 and F2. Only three DMC shared between generations.	[32]

BAT, brown adipose tissue; BuP, butylparaben; BW, body weight; *Cox*2, cytochrome c oxidase subunit II; DDT, dichlorodiphenyltrichloroethane; DMC, differentially methylated CpGs; DMR, differentially methylated regions; DMSO, dimethyl sulfoxide; *fas*, TNF receptor superfamily member 6; HFD, high-fat diet; i.p., intraperitoneal; iAs, inorganic arsenic; PAH, polycyclic aromatic hydrocarbon; PND, postnatal day; POMC, proopiomelanocortin; *Pparγ*, peroxisome proliferator-activated receptor gamma; STD, standard diet; TBT, tributyltin; and WAT, white adipose tissue; ↑, increase; ↓ decrease.

**Table 3 ijerph-20-04711-t003:** Epidemiological studies addressing epigenomic changes related to childhood obesity development after maternal exposure to obesogens.

Compound	Cohort and Sample Size	Objective	Measures	Outcomes	References
Bisphenol A (BPA)	Mother–child pairs (LINA mother–child-study, N = 420). (#046–2006, #206–12- 02072012, University of Leipzig).	To analyze epigenetic alterations in the cord blood of BPA prenatally exposed children and their potential link to overweight development.	BPA concentration: urine from gestating women (week 34). DNA methylation: cord blood. Infant’s follow-up: 1 and 6 years.	↓ Methylation of CpG (cg17580798) in the *MEST* promoter. ↓ Methylation of cg23117250 (*RAB40B*). ↑ *MEST* mRNA levels. BPA prenatal exposure related to longitudinal weight development.	[22]
Bisphenol A (BPA)	Children exposed prenatally to low or high BPA levels based on 80th percentile of maternal BPA levels (N = 59). (IRB No. 1201-010-392).	To identify differentially methylated CpG sites due to prenatal BPA exposure.	BPA concentration: urine from gestating women collected during the second trimester of pregnancy. DNA methylation: infants’ whole blood at 2 and 6 years old. Infants’ follow-up: 2, 4, 6, and 8 years old.	↑ Methylation in cg19196862 (*IGF2R*) associate with ↑ BMI at 2 years of age. ↑ BMI during 4–8 years of age associated with hypermethylation in cg19196862 in girls. ↑ Methylation at cg19249811 (*SVIL*) not associated with BMI.	[33]
Di-2-ethylhexyl phthalate (DEHP)	Mother–child pairs (Hokkaido study) (N = 203) (reference no. 14, 22 March 2012, Hokkaido University Center for Environmental and Health Sciences).	To elucidate the relation between prenatal DEHP exposure and cord blood DNA methylation, as well as the association between DNA methylation and ponderal index (PI) at birth.	Mono(2-theylhexyl)phthalate concentration (MEHP) as indicator of DEHP exposure: maternal blood samples. DNA methylation: cord blood.	Maternal MEHP levels positively correlated to methylation levels in CpG located at 200 bases from the transcription start with of *ZC3H10* (cg26409978) and another mapped to *SDK1* (cg00564857). Enrichment of metabolic pathways, MAPK, Notch, and GnRH signaling pathways, renin secretion, and cortisol synthesis and secretion. ↑ Methylation levels at cg27433759 (*PIK3CG*), cg10548708 (*ACAA1*), and cg07002201 (*FUT9*) related to high levels of MEHP and lower PI.	[34]
Parabens	Mother–newborn pairs (ENVIRONAGE cohort, N = 229) (reference no. B371201216090 and B371201524537).	To determine the association between placental paraben levels and cord blood metabolic biomarkers, epigenetic alterations, and childhood trajectories of BMI z-scores.	Parabens concentrations (methyl (MeP), ethyl (EtP), propyl (PrP and butyl (BuP) parabens): placenta. DNA methylation: cord blood. Infants’ follow up: up to 29 months after birth.	Correlation between higher levels of EtP and hypermethylation of cg08612779 (annotated to *GGT7*). EtP related to decreased longitudinal BMI z-scores.	[35]

*ACAA1*, acetyl-CoA acyltransferase 1; BMI, body mass index; BPA, bisphenol A; BuP, butylparaben; DEHP, diethylhexyl phthalate; EtP, ethylparaben; *FUT9*, fucosyltransferase; GGT7, gamma-glutamyltransferase 7; GnRH, gonadotropin-releasing hormone; MAPK, mitogen-activated protein kinase; MEHP, mono(2-theylhexyl)phthalate; MeP, methylparaben; *MEST*, mesoderm-specific transcript; PI, ponderal index; *PIK3C*G, phosphatidylinositol-4,5-bisphosphate 3-kinase catalytic subunit gamma; *SDK1*, sidekick cell adhesion molecule 1; *SVIL*, supervillin; and *ZC3H10*, zinc finger CCH-type containing 10; ↑, increase; ↓ decrease.

## Data Availability

Not applicable.

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
