# Peer review of "Inherited Epigenetic Hallmarks of Childhood Obesity Derived from Prenatal Exposure to Obesogens"

_ijerph, 2023, doi:10.3390/ijerph20064711_

Round 1

Reviewer 1 Report

The objective of this review was to summarize the current knowledge of epigenetic modifications due to maternal exposure to those obesogens during pregnancy and their potential implication on long-term obesity development in the offspring and transgenerational inheritance of epiphenotypes. The manuscript is well written and organized with a very clear language. It is related to an important topic and new area of research. Only few minor suggestions are below:

1.       Give a recommendation statement at the end of abstract

2.       L165-166: re-write for clarity

3.       L218: I suggest to add sub-section to add other non-persistent organic pollutants

4.       L303: deleted "however"

Author Response

Reviewer 1:

The objective of this review was to summarize the current knowledge of epigenetic modifications due to maternal exposure to those obesogens during pregnancy and their potential implication on long-term obesity development in the offspring and transgenerational inheritance of epiphenotypes. The manuscript is well written and organized with a very clear language. It is related to an important topic and new area of research. Only few minor suggestions are below:

We thank the reviewer for his/her positive appreciation of our work and for his/her suggestions and comments that have helped us to improve the manuscript.

  1. Give a recommendation statement at the end of abstract

We thank the reviewer for this comment. We have updated the abstract according to his/her suggestion.

Lines 23-34

“Abstract: Childhood obesity has reached epidemic levels in developed countries and is becoming a major cause for concern in the developing world. The causes of childhood obesity are complex and multifactorial, involving the interaction between individual genetics and environmental and developmental factors. Among the environmental factors, there is a growing interest in understanding the possible relationship between the so-called environmental obesogens and the development of obesity in children. Exposure to these obesogens such as phthalates, bisphenol A, or parabens, has been identified to promote obesity through different mechanisms such as alteration in adipocyte development from mesenchymal progenitors, interference with hormone receptors, and induced inflammation. However, less attention has been paid to inheritance of epigenetic modifications due to maternal exposure to these compounds during pregnancy. Thus, the aim of this review is to summarize the current knowledge of epigenetic modifications due to maternal exposure to those obesogens during pregnancy as well as their potential implication on long-term obesity development in the offspring and transgenerational inheritance of epiphenotypes.”

  1. L165-166: re-write for clarity

We thank the reviewer for the comment. We have now rewritten the indicated sentence as follows:

Lines 177-178.

“The authors were able to stablish a relationship between increased methylation at cg19196862 (IGF2R) and the BMI at 2 years of age, although such association was lost at later ages.”

  1. L218: I suggest to add sub-section to add other non-persistent organic pollutants

We thank the reviewer for his/her suggestion. We acknowledge that there are other pollutants such as organophosphate pesticides (chlorpyrifos), particulate matter, or perfluorooctanoic acid that have been described to exert obesogenic effects on the progeny after maternal exposure (see PMID: 18166376, 25172901, 31874421). However, to the best of our knowledge, there are no detailed studies focused on transgenerational inheritance of obesity epiphenotypes after maternal exposure. As this review aims to delve into current knowledge on such inherited epigenetic hallmarks of obesity, we have included only those pollutants addressing this topic. The lack of available information on other pollutants prevents us to include another section.   

  1. L303: deleted "however"

Thanks for your suggestion. We have removed it from the revised version of the manuscript.

“Although prenatal PAH exposure has been related with childhood obesity development [87, 88], mechanistic studies are still limited.

Reviewer 2 Report

The review article, "Inherited Epigenetic Hallmarks of Childhood Obesity Derived from Prenatal Exposure to Obesogens" is a clear, well-written manuscript on an often underappreciated subject matter which may have the potential to largely impact obesity clinically.  It is recommended for the authors to address the following critiques below to further enhance this review article:

*Please omit keywords that are already within the manuscript title and replace these words with more appropriates ones.

*It is suggested to compose an additional figure to illustrate the concepts associated with lines 55-68 to thoroughly breakdown these complicated topics.

*A section within the review should focus on discussing the fetal origins hypothesis in the context into how obesogens may play in role within this hypothesis, with appropriate citations referenced.

*Table 1 is very detailed and appreciated, but rather dense in its current format.  It is recommended to better categorize this information by bifurcating this Table into two separate Tables, with examples of separation being rodents used, biological sex of mice, types of mice, etc.  However, the separation of newly constructed Tables by subject/topic should be at the authors' discretion.

*In addition to the prior comment, it it suggested for the authors provide more detail on the limitations of animal models listed in their review article.  While it is acknowledged that this glossed over in lines 339-341, more insight into why rodent models are poor models for obesogens and how extrapolation of the animal study data should be proceeded with caution in relevance to humans should be explicitly stated.

Author Response

Reviewer 2:

The review article, "Inherited Epigenetic Hallmarks of Childhood Obesity Derived from Prenatal Exposure to Obesogens" is a clear, well-written manuscript on an often underappreciated subject matter which may have the potential to largely impact obesity clinically.  It is recommended for the authors to address the following critiques below to further enhance this review article:

We thank the reviewer for his/her positive consideration about our work and agree with him/her regarding the high relevance of the matter addressed in it. We are also thankful for the reviewer’s suggestions and comments, which have helped us to improve the manuscript.

*Please omit keywords that are already within the manuscript title and replace these words with more appropriates ones.

We thank the reviewer for this recommendation. We have now changed the keywords as follows:

Lines 35-36

“Keywords: Obesogen; epiphenotypes; non-persistent organic pollutants; persistent organic pollutants; inorganic arsenic; childhood obesity; endocrine disrupting chemicals.

*It is suggested to compose an additional figure to illustrate the concepts associated with lines 55-68 to thoroughly breakdown these complicated topics.

We thank the reviewer for his/her suggestion. We agree that a figure illustrating the mentioned concepts contained in such lines would be of great interest; therefore, such new figure has been now included in the revised version of the manuscript (new Figure 1).

*A section within the review should focus on discussing the fetal origins hypothesis in the context into how obesogens may play in role within this hypothesis, with appropriate citations referenced.

Thanks for your suggestion. We agree that the fetal origins hypothesis is an interesting topic that should be taken into consideration. However, this topic has been already addressed in other works (see PMID: 21684471 and DOI:10.1098/rstb.2018.0123), which have been properly cited in the revised version of the review as references 18 and 19. In any case, we have now included a paragraph addressing it, as suggested:

Lines 91-97

This agrees with the fetal origins of health and disease hypothesis, which states that the risk for the development of long-term diseases is associated with events occurred during early developmental periods such as maternal exposure to environmental toxins [18]. For instance, it has been demonstrated that maternal smoking during pregnancy can lead to decreased birth weight and increased risk of obesity development during childhood [19]. Thus, due to their ability to promote epigenetic dysregulation on exposed individuals, obesogens have aroused lately the interest of researchers. Furthermore, prenatal exposure to these obesogens can cause epigenetic modification that may produce adverse phenotypic outcomes in adulthood and even be transferred to subsequent generations [20].”

*Table 1 is very detailed and appreciated, but rather dense in its current format.  It is recommended to better categorize this information by bifurcating this Table into two separate Tables, with examples of separation being rodents used, biological sex of mice, types of mice, etc.  However, the separation of newly constructed Tables by subject/topic should be at the authors' discretion.

Thanks for this remark. We agree with the fact that the original Table 1 could be quite dense for the reader. Therefore, we have split it into two different ones for clarity purposes as follows:

Table 1. Animal studies on transgenerational inheritance of obesity epiphenotypes after ancestral exposure to obesogenic non-persistent pollutants.

Table 2. Animal studies on transgenerational inheritance of obesity epiphenotypes after ancestral exposure to obesogenic persistent pollutants.

 *In addition to the prior comment, it it suggested for the authors provide more detail on the limitations of animal models listed in their review article.  While it is acknowledged that this glossed over in lines 339-341, more insight into why rodent models are poor models for obesogens and how extrapolation of the animal study data should be proceeded with caution in relevance to humans should be explicitly stated.

Thanks for the suggestion. We have now added a statement regarding the limitations of animal studies in section 5.

Lines 351-354

However, although the use of these models is fundamental to understand the mechanisms underlying the transgenerational inheritance of epiphenotypes, results should be extrapolated with caution in relation to humans.”

Reviewer 3 Report

The authors summarized the current knowledge of epigenetic modifications due to maternal exposure to those obesogens during pregnancy as well as their potential implication on long-term obesity development in the offspring and transgenerational inheritance of epiphenotypes. There are several problems:

1.      In the abstract, the authors use too many words to introduce the background and do not describe their contributions clearly.

2.      It is known that obesity is associated with various diseases (PMID: 29258237). The authors should also include obesity-related diseases and consider their effects.

3.      Figure 1. Schematic effect of obesogens on epigenetic hallmarks leading to childhood obesity, for the two DNA strains, could the authors add the molecular mechanism names or key genes/pathways? It will be more informative for the readers.

4.      The authors only discussed the effects of the mother's side. How will the father's side affect childhood obesity?

5.      The authors need to add a summary mechanism figure incorporating the results from Table 1 and Table 2.

Author Response

Reviewer 3:

The authors summarized the current knowledge of epigenetic modifications due to maternal exposure to those obesogens during pregnancy as well as their potential implication on long-term obesity development in the offspring and transgenerational inheritance of epiphenotypes. There are several problems:

  1. In the abstract, the authors use too many words to introduce the background and do not describe their contributions clearly.

We thank the reviewer for this comment. Following reviewer’s suggestion, we have updated the abstract in the revised version of the manuscript as follows:

Lines 23-34

“Abstract: Childhood obesity has reached epidemic levels in developed countries and is becoming a major cause for concern in the developing world. The causes of childhood obesity are complex and multifactorial, involving the interaction between individual genetics and environmental and developmental factors. Among the environmental factors, there is a growing interest in understanding the possible relationship between the so-called environmental obesogens and the development of obesity in children. Exposure to these obesogens such as phthalates, bisphenol A, or parabens, has been identified to promote obesity through different mechanisms such as alteration in adipocyte development from mesenchymal progenitors, interference with hormone receptors, and induced inflammation. However, less attention has been paid to inheritance of epigenetic modifications due to maternal exposure to these compounds during pregnancy. Thus, the aim of this review is to summarize the current knowledge of epigenetic modifications due to maternal exposure to those obesogens during pregnancy as well as their potential implication on long-term obesity development in the offspring and transgenerational inheritance of epiphenotypes.”

  1. It is known that obesity is associated with various diseases (PMID: 29258237). The authors should also include obesity-related diseases and consider their effects.

We thank the reviewer for his/her suggestion. We acknowledge that obesity is related with a wide variety of long-term diseases and that obesity-related complications have should not be disregarded. However, the aim of this review was to provide a concise overview of the current available information regarding the epigenetic inheritance of obesity hallmarks in early childhood derived from maternal exposure to environmental obesogens.

Thus, given that most of the obesity-related diseases develop in the long term, being mainly evident during adolescence, we consider that this subject, despite of its great importance, is beyond the scope of this review. Nonetheless, as we agree with the reviewer on the implications that obesity has in other type of diseases beyond metabolic ones, we have modified the introduction to highlight such relationship.

Lines 47-49

“In fact, obese children are more likely to be obese in adulthood and are at greater risk of long-term metabolic diseases including type 2 diabetes mellitus (T2DM), hypertension, nonalcoholic fatty liver disease (NAFLD), sleep apnea obstructive sleep (SAOS) anddyslipidemia [2]. Moreover, obesity has also been suggested to have a role in the development of other health conditions such as psychiatric- or bone-related diseases [3].”

[3] Chen L, Zhang YH, Li J, Wang S, Zhang Y, Huang T, et al. Deciphering the Relationship between Obesity and Various Diseases from a Network Perspective. Genes (Basel). 2017;8.

  1. Figure 1. Schematic effect of obesogens on epigenetic hallmarks leading to childhood obesity, for the two DNA strains, could the authors add the molecular mechanism names or key genes/pathways? It will be more informative for the readers.

We thank the reviewer for his/her comment. We have added a new schematic figure illustrating the molecular mechanisms known to be involved in the obesogens effects (new Figure 1), while keeping former Figure 1 as the Figure 2 in the revised version of the manuscript.

  1. The authors only discussed the effects of the mother's side. How will the father's side affect childhood obesity?

Thanks for your interesting observation. Currently there are only very few studies addressing the parental’s contribution to obesity. Indeed, of the few existing studies, none of them are related to parental pre-conceptional exposure to environmental obesogens. However, as we consider that this subject should not be neglected but encouraged in future studies, we have outlined it in the future perspectives and conclusions section:

Lines 354-356

“Therefore, another conclusion is that additional longitudinal studies should be performed not only in wide mother-offspring but also in father-offspring cohorts to study epigenetic modifications caused by obesogens, with a special focus on the time frame including pregnancy and the first 2 years of life after birth.” 

  1. The authors need to add a summary mechanism figure incorporating the results from Table 1 and Table 2.

We thank the reviewer for his/her comment. We have now added a new schematic figure illustrating the molecular mechanisms known to be involved in the obesogens effects (new Figure 1).

Round 2

Reviewer 2 Report

The authors have sufficiently satisfied my reviewer concerns.

Reviewer 3 Report

The authors have answered my questions.